# A Self-Regulated Microfluidic Device with Thermal Bubble Micropumps

**DOI:** 10.3390/mi13101620

**Published:** 2022-09-28

**Authors:** Gang Guo, Xuanye Wu, Demeng Liu, Lingni Liao, Di Zhang, Yi Zhang, Tianjiao Mao, Yuhan He, Peng Huang, Wei Wang, Lin Su, Shuhua Wang, Qi Liu, Xingfeng Ma, Nan Shi, Yimin Guan

**Affiliations:** 1Department of Microelectronics, Shanghai University, Shanghai 200000, China; 2Shanghai Industrial μTechnology Research Institute, Shanghai 200000, China; 3Shanghai Aure Technology limited Company, Shanghai 200000, China; 4Institute of Translational Medicine, Shanghai University, Shanghai 200000, China

**Keywords:** thermal bubble micropump, microfluidics, competitive immunoassay, aflatoxin

## Abstract

Currently, many microchips must rely on an external force (such as syringe pump, electro-hydrodynamic pump, and peristaltic pump, etc.) to control the solution in the microchannels, which probably adds manual operating errors, affects the accuracy of fluid manipulation, and enlarges the noise of signal. In addition, the reasonable integration of micropump and microchip remain the stumbling block for the commercialization of microfluidic technique. To solve those two problems, we designed and fabricated a thermal bubble micropump based on MEMS (micro-electro-mechanical systems) technique. Many parameters (voltage, pulse time, cycle delay time, etc.) affecting the performance of this micropump were explored in this work. The experimental results showed the flow rate of solution with the assistance of a micropump reached more than 15 μL/min in the optimal condition. Finally, a method about measuring total aflatoxin in Chinese herbs was successfully developed based on the integrated platform contained competitive immunoassay and our micropump-based microfluidics. Additionally, the limit of detection in quantifying total aflatoxin (AF) was 0.0615 pg/mL in this platform. The data indicate this combined technique of biochemical assays and micropump based microchip have huge potential in automatically, rapidly, and sensitively measuring other low concentration of biochemical samples with small volume.

## 1. Introduction

Originally, microfluidics was developed for analyzing tiny volume of samples [1,2], specifically for micro-chromatography applications such as capillary electrophoresis. Microfluidic technique started to widely appear in the research and industrial fields after the Oak Ridge National Laboratory of the United States improved the application of capillary electrophoresis in microchips [3,4,5,6]. However, the essential requirement of external power sources and analysis equipment make the whole microchip system complicated and hinder its wide usage. To address the problem about power system of microchips, the micropump is developed, which has the characteristics of small size, light weight, and high integration. Normally, micropumps can be divided into two main types: mechanical micropump and non-mechanical micropump [7]. In comparison to mechanical micropumps, non-mechanical micropumps are more popular because they are convenient for integration. Zebing et al. successfully proposed electrohydrodynamic (EHD) pumps and the soft EHD rollers with simple structures and used those two devices to achieve pushing fluids forward and even bidirectional flow [8,9]. In addition, thermal bubble micropump is another important branch of the non-mechanical micropump, which utilizes thermal bubble to push liquid moving forward according to orders from controller.

Thermal bubble technology [10,11,12] has been used in many research experiments and industries [13] since it was firstly applied in the inkjet [14,15,16]. Specially, the thermal-bubble-based microvalves and micropump have played quite important roles in different fields, such as aerospace [17], biomedicine [18,19], and chemistry [20,21]. Currently, resistance heating [10,22] is the most conventional and simplest method in bubble generation.

In current microfluidic systems, the sample driving, processing, and detection do require the assistance from external equipment, which probably brings in extra artificial errors and greatly affects the industrial development of microfluidic technology. Shi et al. used a microfluidic device to improve the detection sensitivity of insulin [23]. Their system used LabView codes to control nitrogen to open/close the membrane-based valves for manipulating the fluids in the microchannels. The transportation time of N_2_ from nitrogen tank to valves via long Teflon tubing could be viewed as delay time, which resulted in the tiny deviations of flow rate and droplet sizes, and probably caused the detection accuracy of insulin in this system. Sara Gómez-de Pedro et al. [24] demonstrated an optical microfluidic system with modified gold nanoparticles to monitor Hg(II), and the external force equipment added in certain operational difficulties to operators. To address those problems, we proposed an innovative concept of intelligent microfluidics based on CMOS (complementary metal-oxide-semiconductor)-MEMS (micro-electro-mechanical systems) technology and developed a thermal-bubble-micropump-based microchip with some outstanding characteristics, such as internal power source, small size, high precision, and intelligence.

Compared to traditional thermal bubble micropump, the thermal bubble micropump introduced in this project has smaller size and simpler structure; In terms of performance, each-time-pump volume and flow rate of liquid have improved significantly, and some biochemical analyses also can be accomplished in this integrated microchip. We used this microsystem for rapidly analyzing the aflatoxin content in various traditional Chinese medicine samples. In this work, the accurate measurement of aflatoxin in varieties of Chinese herbal medicines had been achieved. Compared to traditional analysis methodology, the efficiency and limit of detection were improved significantly in this work. At a word, this microdevice is suitable for rapidly screening the quality of Chinese medicine which should help Chinese medicine enterprises to establish a more efficient and stable quality control system.

## 2. Materials

Heat-generating films were obtained from Shanghai Industrial μTechnology Research Institute (Shanghai, China). Flowmeter was purchased from Dalian Leader Fluid Control Technology (Liaoning, China). Thermal grease was acquired from Dongguan City Shun trillion electronic Alwayseal Technology (Guangdong, China). Rapid detection clips were obtained from Yungong Zhida Precision Technology (Shenzhen, China). Soft tubes were bought from Gate Automation (Shanghai, China). Aflatoxin-modified Bovine serum albumin (AF-BSA) and primary antibodies were obtained from Shanghai University of Traditional Chinese Medicine (Shanghai, China). Fluorescently label antibodies were purchased from Cell Signaling Technology (Danvers, MA, USA). Samples (Semen Platycladi, Pericarpium Citri Reticulataeas, Soft-Shelled Turtle, Hellebore, and Semen Coicis) were obtained from Tianjin University of Traditional Chinese Medicine (Tianjin, China). Fifty percent Glutaraldehyde was obtained from Shanghai Aladdin Biochemical Technology (Shanghai, China). PBS and PBST buffer were purchased from Beijing Solarbio Science & Technology (Beijing, China). Aflatoxin was obtained from National Institutes for Drug Control (Beijing, China).

## 3. Results and Discussion

### 3.1. Thermal Bubble Micropump Characteristics

Figure 1a describes the schematic diagram of the micropump’s working principle. When liquid flows into the microchannel from inlets, it is heated by the array heating resistors firstly, and then vaporization phase transition formed above the heating resistors sequentially (Appendix A). Finally, the liquid-pumping is generated by directional compression of liquid in the microchannels. In addition, the position of each heating resistor has important effects on bubble generation and liquid pumping. Based on the theoretical analysis of micropump (see Appendix A) we finally set up 24 side-by-side heating resistors, and the interval distance between each two resistors was 24.3 μm.

The micropump was fabricated in the following steps (Appendix A): Firstly, a row of heat-generating films (resistors) was placed on the silicon substrate, and then two holes were formed on the surface of silicon via dry etching. The microchannel was generated by exposure, development, and curing. Finally, the top of the side wall was covered with dry photoresist film.

Some bubbles would generate in the solid-liquid edge of the inlets when the thermal bubble micropumps worked. To address this issue, three diverse types of inlet structures were designed to see which one would perform better (Appendix A). Our first design was a square with round corners at the right angles. A round hole was arranged at the entrance in the second structure. In the third design, several cylinders were added behind the round hole. In the first structure, the micropump stopped working when the entire microchannel was filled with bubbles. The micropump in the third design was blocked by bubbles more quickly. The second method had the best experimental result among these three methods: bubbles were not easy to accumulate, and the working time was the long enough. In addition, we enlarged the design of the round hole to facilitate the filling efficiency of liquid. Figure 1b,c shows the last version of the micropump structure and the Appendix A demonstrates the performance of our thermal bubble-based micropump.

### 3.2. Micropump Characterization

The generation of pulse voltage difference between two heating resistors will affect the temperature of the resistors. Therefore, it is worth exploring or even quantifying the relationship between electricity and heat if we would like to control these resistors well in the following work. In addition, the working interval time should determine the flow volume and flow rate of the liquid in some ways based on our previous experience. At a word, the pulse voltage, pulse time, and time intervals of electrodes, as well as the number and distance interval of heating resistors will be studied here to demonstrate how they influence the working performance of the micropumps (Figure 2a).

In this system, the working time of the heating resistor is mainly determined by the interval distance between each two adjacent heating resistors and the pulse time of the electrodes. Microchannels will probably be blocked by superfluous bubbles if heat resistors work for too long. Both the number and sizes of bubbles significantly decrease when heat resistors run in a short period, and seriously affects the fluid’s flow rate. Meanwhile, the interval time of the two electrode pulses is also extremely important in determining the working interval of the adjacent heating resistors. The bubbles disappear quickly and liquid flows slowly if the interval time is too short. However, bubbles will disappear slowly and the microchannels probably be choked by bubbles if the interval time is too long. For verifying how those various parameters influence the micropumps and found out the optimal interval time, we adjusted the cycle delay time (T_D_) after all electrodes were heated in each round according to the equation below: $$W=U^{2}/R*t$$
where *R* represents the heating resistance value in this equation, *U* is the pulse voltage applied to electrodes, and t represents low pulse time (T_L_) or high pulse time (T_H_). Figure 2 represented the effects of high/low pulse time and cycle delay time on the flow rate, and flow meter was used to measure the flow rates at room temperature. In Figure 2b, the P-values were 0.121 and 0.175, which were bigger than 0.05 when T_L_ = 650 ns, 750 ns, and it meant the different high pulse time (T_H_) did not obviously affect the flow rate. However, the T_H_ obviously affected the flow rate when T_L_ = 850 ns and 950 ns. There was no significant difference in the flow rate when T_H_ = 750, 850, 950 ns no matter the T_L_ values, but the significant difference existed when T_H_ = 650 ns (P = 0.0045). In Figure 2c, we mainly explored the effect of cycle delay time (T_D_) on the flow rate, and we concluded T_D_ (2400, 3600, 4800, 6000 ns) did not have obvious effects on the flow rate based on the analysis of variance (T_H_ = 650, 750, 850, 950 ns, T_L_ = 650 ns).

The stability of micropumps inserted in microchips must be completely verified before usage. As mentioned above, the micropump is primarily excited by heating resistors, and bubbles start to aggregate at the inlet of the micropump when the temperature of resistors is high enough. Air will suck into the micropump if the volumes of bubbles are too large, which probably results in the micropump not working. Therefore, we speculate that the temperature should have an important effect on the working condition of micropump. To accurately measure the temperature of the microchips, we used a thermal imager to record and analyze the real-time temperature and temperature distribution around the microchips. Eventually, we found the temperature of the regions which were close to the heating plate was about 66 °C and the temperature of regions, which were relatively far away from the heating plate, were around 55 °C (Appendix A).

To figure out the optimal working temperature of the micropump, we inspected the performance of micropumps in three different conditions (thermal-grease, water-cooling system, and room temperature) (Appendix A). The results indicated the pumped volume of liquid exceeded 225 µL in both of water-cooling and thermal-grease conditions, but the pumped volume was less than 150 µL at room temperature within same time. Therefore, precise control of temperature is essential in enhancing the stability of the micropump. In addition, we tried to adjust the interval time between each two adjacent cycles to avoid the excessive accumulation of thermal energy inside micropumps. The Figure 3b also showed that the continuous pumping volume of the micropump almost keep relatively consistent during its running. In addition, we tried to apply the specific working mode (work-pause-work) to maintain a relatively low temperature for lifting micropump’s working stability. Water cooling was the most effective method for controlling the temperature of the thermal bubble micropump in these three methods (Figure 3a,b).

The basic working principle of thermal bubble micropump is that each heating resistor works continuously and cyclically to generate thermal bubbles to push liquid forward. Here, we explored the relationship between the pump fluid volume and cycle delay time, as well as the intrinsic interaction between flow rate and cycle delay time. In each graph, three color curves represent the results of different test samples. It is obvious that the liquid flow rate gradually declined from 11 µL/min to 6 µL/min, and the volume achieved to ~66 µL when cycle delays changed from 10 µs to 35 µs (Figure 3c,d). The *p*-value of flow rate = 0.0262 < 0.05, so there was significant difference in the flow rates of various delay times. The *p*-value of pump volume = 0.169 > 0.05, which meant there is no significant difference in the pump volumes of various delay times. However, the standard deviations were not low enough in some groups, and it represented the stability of micropump system still required improving when the thermal bubble micropump must work for long time. In addition, some supplemental description was necessary to add in the analysis of Figure 3c,d. We normally calculate the volume with the equation: Volume = Flow rate * Flow time, and the premise is that the micropump works stably during the measured time. However, the short cycle delay time still rapidly increased the temperature of micropump and micropump stopped working though in the water-cooling condition. Therefore, the micropump had fast flow rate in the short cycle delay time, but the real pump-time was short, and the final pump volume was low; The flow rate of the micropump was slower when cycle delay time was longer, but it pumped for a much longer time and the final volume was relatively higher.

### 3.3. Active Microchip with Micropumps

Our group designed an automatic microfluidic chip coupled with micropumps for biochemical analysis. The microchip mainly consists of three parts: cover plate, microfluidic channels, and analysis platform, which has a water-cooling section, reaction chamber, samples inlets, waste collection tank, ventilation hole, and connecting channel. The structure of this microchip is shown in Figure 4. To well maintain the temperature of the micropump, a water-cooling system was added to the cover plate area in the microfluidic chip, as shown in Figure 4a. Considering various parameters (such as viscosity, surface tension, boiling point, etc.), the spacings and sizes of liquid reservoirs and micro-channel widths were determined and designed after preliminary simulation. To satisfy the requirements of biochemical reactions, three sample reservoirs were designed to avoid the cross-contamination of reagents and prohibit side reactions, which were caused by reactants mixing in advance. Another specific component was added to remove air bubbles at the front end of the reaction zone which greatly avoids the generation of excessive bubbles, as shown in Figure 4b,d,e. Figure 4e exhibits the final version of our integrated microchip.

## 4. Immunoassays in Active Microfluidic Chip

Microfluidic devices are normally used for precisely controlling and manipulating liquid in micro or nano-size channels, which have the potential to be platforms with the properties of miniaturization, integration, and automation. In this work, an immunofluorescence detection method for total aflatoxin (AF) was developed based on a microfluidic chip (Figure 5a). This system mainly contains four parts: microchip, thermal bubble pumps, modified silicon-chip, and optical equipment. The whole reaction took place on the surface of silicon chip, which was inserted in the wide chamber in the microchip, and the reagents could be categorized into aflatoxin-modified Bovine serum albumin (AF-BSA), primary antibodies, samples, and secondary antibodies. The thermal bubble pump was used to transport reagents dissolved in aqueous solution and wash away unreacted items in this work (Figure 5b).

The silicon-based protein microchip on this system worked as a “reaction zone”, which was modified in advance. A 300-nm depth of oxide layer was coated on the surface of silicon slice firstly, and another ammonia-base layer was fabricated on the top of oxide layer. Then, the silicon slice was activated with glutaraldehyde at 37 °C under alkaline conditions. Finally, the AF-BSA were sprayed onto the surface of the silicon chip via special inkjet printer, and the modified silicon chip was inserted into the microchip.

After accomplishing the fabrication of the microchip, the labeled secondary antibodies, mixture of primary antibodies and sample, and PBST wash buffer were injected into three different reservoirs in the microdevice. At the beginning, the mixture of the primary antibodies and samples were transported into the reaction zone via pump and incubation. During the incubation, some primary antibodies bind with target samples and a few antibodies were well captured by the AF-BSA on the surface of silicon chip, and the unreacted primary antibodies were washed away from reaction zone. Then, the labeled secondary antibodies flowed into the reaction zone and specifically recognized the primary antibodies trapped in the detection area. Finally, washing away unreacted secondary antibodies, quantifying fluorescence signals, and calculating the concentration of analytes (Figure 5c). The limit of detection of total aflatoxin (AF) was 0.0615 pg/mL in this microchip, which was significantly lower than the global minimum detection standard.

To validate the measurement accuracy of our microchip, the aflatoxin concentrations of some herbal samples were quantified with both of this highly automatic microfluidic system and traditional high-performance liquid chromatography (HPLC). A total of 0.25 pg/mL and 0.15 pg/mL of aflatoxin were added into Semen Platycladi and Pericarpium Citri Reticulate as controls, respectively. In this work, the detection values of samples (Semen Platycladi + 0.25 pg/mL aflatoxin) from microchip and HPLC were 0.2470 pg/mL and 0.2532 pg/mL, respectively. In addition, the recoveries of aflatoxin were 98% in our microchip and 101% in HPLC. Comprehensively considering the results about quantifying aflatoxin in the four herbal samples (Semen Platycladi, Pericarpium Citri Reticulataeas, Hellebore, and Semen Coicis) (Table 1), it was concluded that our integrated microchip did have higher precision than HPLC in analyzing aflatoxins. Such a system would be ideal to apply to quantify low concentration or volume of samples with corresponding biochemical assays.

## 5. Discussion

In this project, our group developed the thermal bubble-based micropump which has the characteristics of high flow rate and large pumping volume. Pulse voltage, pulse time, temperature, and the interval of pulse time have essential effects on the working performance of the micropump. What is more, three temperature controlling methods were tested in this work and water-cooled method was viewed as the best techniques among them.

Microfluidics is a rapidly developing Microanalysis technology. We invented a system for total aflatoxin measurement on the platform of micropump-based microchip. Compared to the traditional ELISA, this integrated system has several advantages: (1) The automation of sample analysis in our microchip is much higher than traditional detection methods, so it requires less labor and eliminate the error brought from workers; (2) It has high detection sensitivity which reaches 0.0615 pg/mL; (3) It has the potential in simultaneously analyzing multiple samples with improved microchip: (4) The volume of reagents is as low as ~10 µL. As a word, thermal bubble-driven microfluidic chips have broad application prospects though there are some holdbacks in this road.

## Figures and Tables

**Figure 1 micromachines-13-01620-f001:**
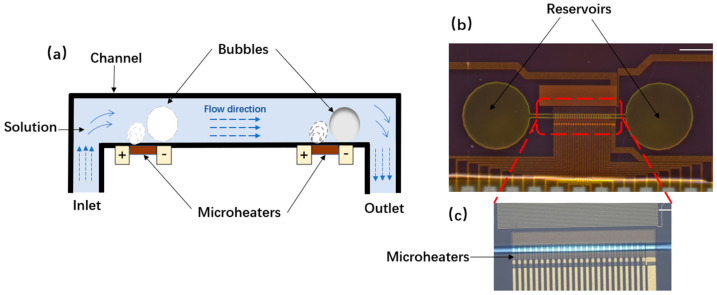
(**a**) Schematic diagram of the micropump. (**b**) Entity graph of heating resistors and micropump structure and the (Scale bar: 0.5 μm). (**c**) The amplified photo of microheater (Scale bar: 0.07 μm).

**Figure 2 micromachines-13-01620-f002:**
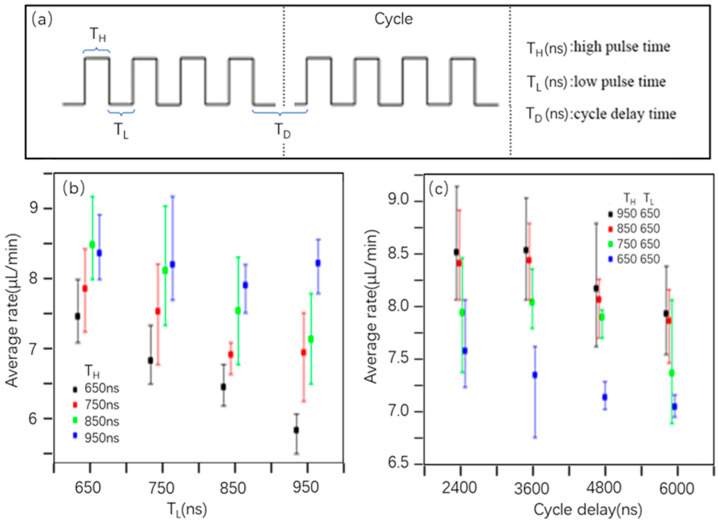
(**a**) The driving waveform imposed on the heater. (**b**) The relationship between average flow rate and low pulse time (T_L_), high pulse time (T_H_). (**c**) The effect of cycle delay time (T_D_) on the flow rate.

**Figure 3 micromachines-13-01620-f003:**
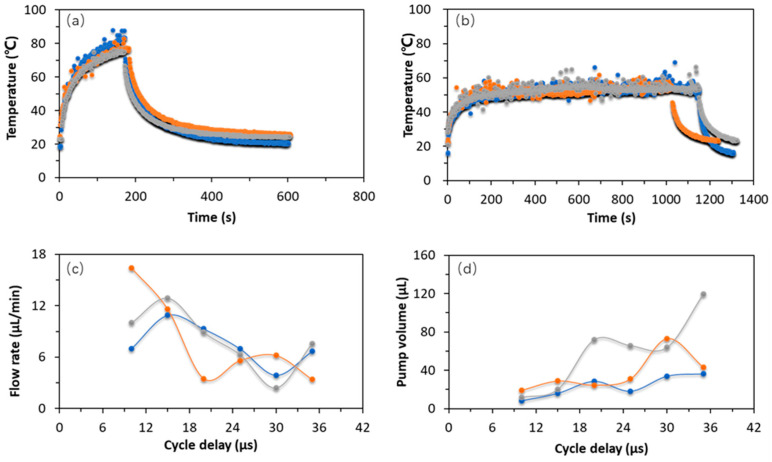
(**a**) Temperature curves of micropumps at room temperature. (**b**) Temperature of micropump under water cooling condition. (**c**) The effects of cycle delay time on flow rate. (**d**) The relationship between pump volume and cycle delay time. (Three colors represent three replicates).

**Figure 4 micromachines-13-01620-f004:**
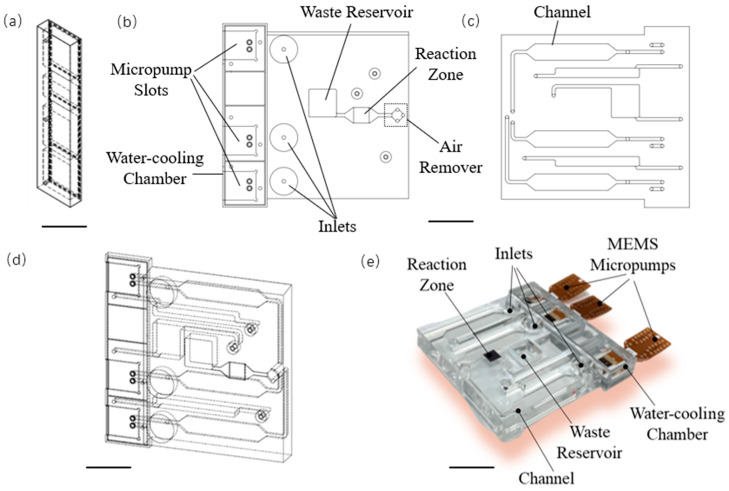
(**a**) The layout of cover plate for the micropump. (**b**,**c**) Both sides of microchip. (**d**) The 3D design sketch of the microchip without micropump. (**e**) The integrated version of the micropump and the microchip. (Scale bar: 24mm).

**Figure 5 micromachines-13-01620-f005:**
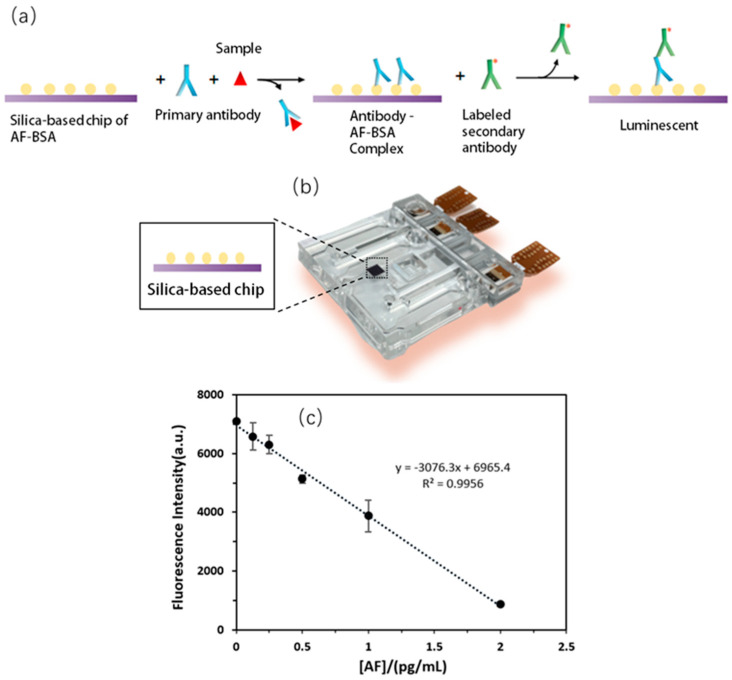
(**a**) Schematic diagram of the principle of competitive immunoassay. (**b**) The modified silicon-based chip in the reaction chamber of microfluidic device. (**c**) The calibration curve of aflatoxin measurement, which was fitted to linear model, where R^2^ = 0.9956.

**Table 1 micromachines-13-01620-t001:** Measurements of total aflatoxin in samples from microchip and HPLC.

Sample	Microchip (pg/mL)	HPLC (pg/mL)
Semen Platycladi	NA ^a^	<0.0500 *
Semen Platycladi + 0.25 pg/mL AF	0.2470	0.2532
Pericarpium Citri Reticulataeas	NA ^a^	<0.0500 *
Pericarpium Citri Reticulataeas + 0.15 pg/mL AF	0.1641	0.2000
Soft-Shelled Turtle	NA ^a^	<0.0500 *
Hellebore	0.0697	<0.0500 *
Semen Coicis	NA ^a^	0.0512

^a^ NA means out of detection range; * The LOD of the aflatoxin in HPLC is 0.0500 pg/mL.

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
