# Peer review of "A Self-Regulated Microfluidic Device with Thermal Bubble Micropumps"

_micromachines, 2022, doi:10.3390/mi13101620_

Round 1

Reviewer 1 Report

The authors describe thermal micropumps to drive fluid flow in microfluidic devices. The manuscript shows the optimisation of device design and operational parameters (including voltage, pulse time, and delay time) and validation of the micropump in a functional assay for the detection of aflatoxin.

In my view the manuscript does not show enough novelty and does not offer enough detail to be accepted for publication. I also question the choice of developing a bubble-based micropump to be used in microfluidic devices given the issues typically associated with the occurrence of bubbles in microfluidics. The manuscript is mostly focused on the optimisation of the pump working parameters, which in my view are not sufficient for this publication, and would be better suited as electronic supplementary information of a manuscript that uses the pumps on a functional assay with high applicability.

Some other issues include:

In the introduction, references 15 to 23 do not appear in the text.

Also in the introduction, I cannot find in refs 30 and 31 the issues mentioned in this manuscript. Actually, the authors of article 31 mention that their system is ‘of easy operation’. In article 31, I don’t see where the use of external force sources can lead to significant delays. Also, what is meant by external force source? Flow-driving mechanisms such as syringe pumps, peristaltic pumps or pressure controllers?

What do the different colours mean in Fig. 2d?

Th appears in the text as an acronym before being introduced in full.

There is very little difference in flow rate when different pulse times and cycle delays are used. Also with large stdev. Does it actually make much difference?

Fig. 3 c and d show high variability in the flow rate and pump volume with no clear trend for different cycle delays. Could it be due to experimental errors? Or perhaps with the error associated with the measurement? How was flow rate measured?

Figure 4 should be improved, include a more detailed legend and also show a 2D version for better clarity.

Figure 5c should be improved with better formatting. How many times was the competitive immunoassay tested? What is the N number of this experiment and standard deviation?

If Pericarpoium Citri and soft-shelled turtle were out of detection range, how was semen coices quantified as 0 pg/mL?

Other minor issues (there are more examples):

Line 13: ‘affecting’ or ‘that affect’

Line 15: rate of ‘aqueous’? Aqueous solution? Water?

Line 27: ‘Until’ does not make sense here

Line 30: complicated, not complicate

Line 35: In what terms are non-mechanical micropumps more biocompatible?

Line 37: what does ‘according to codes’ mean?

Line 44:  processing, not procession

Line 45: artificial, not inartificial

Line 49: should have

Line 88: it is heated

Lines 106/107: The first design was… -> this sentence does not make sense. Also, ‘angles’ not ‘angels’.

Line 117: It shouldn’t be assumed that ‘we all know’.

Line 180: Approved by whom?

Line 205: Which specific component?

Author Response

Thanks for your so helpful suggestions. We have improved a lot based on your advice. Please see the attached report. 

Reviewer 2 Report

The authors developed the thermal bubble-based micropump which has the characteristics of high flow rate and large pumping volume. Pulse voltage, pulse time, temperature, and pulse time interval have essential effects on the operating performance of the micropump. They also proposed three temperature-controlling methods in their work. In conclusion, their pump shows a potential pump in some fields like microfluidic chips. 

1. In the introduction, some non-mechanical micropumps should be considered. like some researchers using direct-electrical driven methods, for example, a Fluidic rolling robot using voltage-driven oscillating liquid and a Bidirectional electrohydrodynamic pump with high symmetrical performance and its application to a tube actuator. These contents should be included in their introduction. 

2. The authors did an excellent job with the Thermal Bubble Micropump. There is a small question, how did they control the generation of bubbles at each time? In figure.1a, how to define the inlet and outlet since the flow direction is unknown as the bubble generates. 

3. In figure2, they show the interaction between average flow rate and other various parameters. This is interesting since the response time is short. The small thing is that why do they need a cycle time delay? and how about the impact of duty on their devices?

4. The authors show that water cooling was approved to be an effective method for controlling the temperature of the thermal bubble micropump in figure 3. The results are impressive. There are no legends for each figure and please show them. 

5. The application of their pumps is a nice direction for exploiting the new field for a pump. Such a system would be ideal to apply to quantify low concentration or volume of samples with corresponding biochemical assays. I have a suggestion, they should study the pressure performance of this pump or power generation if possible.

Author Response

Thanks for your positive comments on our work and we also improved our manuscript based on your suggestions. The report is attached below.

Reviewer 3 Report

A Self-Regulated Microfluidic Device with Thermal Bubble Micropumps

Overall the paper is of decent quality. There are multiple places where the authors can improve the language by being more precise. It would also be better if the authors could clarify the application and purpose of the system. 

Specific comments below: 

  1. Line 11. "To solve current problems (external force driving and integration) in microfluidic..." This sentence is not written clearly. What is the current problem? 
  2. Line 102 "The inlet needs to be pre-filled with liquid firstly, and bubbles easily generated 102 around the inlet due to surface energy". This sentence is another example of language improvements needed. 
  3. Figure 1 - not clear what the illustration means. 
  4. Figure 2 - what does the data mean? The authors should also provide theoretical calculations or expectations of these values to compare the measurements to, 
  5. Line 163, the opening sentence is unclear. 
  6. Figure 3: the authors need to provide some statistical analysis among the three replicates to conclude whether it is reproducible. Also, it is interesting to explain why the flow rate and volume vary so much. 
  7. Figure 5. could the author explain a little why this is an "immunoassay"? 

Author Response

Thanks for your advice. We have modified our paper to your comments and hope the attached report can help answer your questions.

Round 2

Reviewer 1 Report

There is very little difference in flow rate when different pulse times and cycle delays are used. Also, with large stdev. Does it actually make much difference?

Answer: That is good question. We have done the analysis of significant difference and made up some conclusions in the paper:

The data represented the effects of high/low pulse time and cycle delay time on the flow rate in Figure 2. In the Figure 2b, the P-values were 0.121and 0.175 which were bigger than 0.05 when TL (low pulse time) = 650 ns, 750 ns, and it meant the different high pulse time (TH) did not obviously affect the flow rate. However, the TH obviously affected the flow rate when TL = 850 ns and 950 ns. There was no significant difference in the flow rate when TH = 750, 850, 950 ns, no matter the TL values, but the significant difference existed when TH = 650 ns (P= 0.0045). In the Figure 2c, we mainly explored the effect of cycle delay time (TD) on the flow rate. And we concluded TD (2400, 3600, 4800, 6000 ns) did not have obvious effects on the flow rate based on the analysis of variance (TH = 650, 750, 850, 950 ns, TL= 650 ns).

This was precisely my point. Is all this investigation relevant? We are talking about changes of 7 to 8.5 uL/min (maximum). 

Fig. 3 c and d show high variability in the flow rate and pump volume with no clear trend for different cycle delays. Could it be due to experimental errors? Or perhaps with the error associated with the measurement? How was flow rate measured?

Answer: As we talked in the paper, the working stability is the key limitation of thermal bubble micropumps because it is difficult to maintain temperature accurately for long time. The stability of the micropump decreased after long-time working which leaded to the data varied highly here. Though we selected the water-cooling method and try to overcome this issue, there is some room to improve in our future work. We bought a microfluidics low-flow liquid flow meter from ELVE FLOW and used it to measure the flow rate.

The use of the Elveflow flow meter should be mentioned.

The temperature is actually quite stable under water-cooling conditions - at around 50 degrees Celsius. Yet flow rates range from 3 to almost 18 uL/min and the volume pumped from 10 to 120 uL/min. These are massive differences. The standard deviations must be huge.. I do think you need to do some more repetitions of these experiments. Also, how come does the flow rate decrease but the volume pumped increases?

Author Response

Many thanks for your impressive suggestions, we have answered your questions and improved our manuscript again.

Reviewer 2 Report

The authors carefully dealt with my questions. 

Author Response

Sounds great. Appreciate your useful advice again.